# Licochalcone D Ameliorates Oxidative Stress-Induced Senescence via AMPK Activation

**DOI:** 10.3390/ijms22147324

**Published:** 2021-07-07

**Authors:** Nagarajan Maharajan, Chitra Devi Ganesan, Changjong Moon, Chul-Ho Jang, Won-Keun Oh, Gwang-Won Cho

**Affiliations:** 1Department of Biology, College of Natural Sciences, Chosun University, 309 Pilmun-daero, Dong-gu, Gwangju 501759, Korea; geneticnaga1990@gmail.com (N.M.); gchitradevi8591@gmail.com (C.D.G.); 2BK21-Plus Research Team for Bioactive Control Technology, Department of Life Science, Chosun University, Gwangju 61452, Korea; 3Department of Veterinary Anatomy, College of Veterinary Medicine and BK21 FOUR Program, Chonnam National University, Gwangju 61186, Korea; moonc@chonnam.ac.kr; 4Department of Otolaryngology, Chonnam National University Medical School, Gwangju 61469, Korea; chulsavio@hanmail.net; 5Korea Bioactive Natural Material Bank, Research Institute of Pharmaceutical Sciences, College of Pharmacy, Seoul National University, Seoul 08826, Korea; wkoh1@snu.ac.kr

**Keywords:** licochalcone D, oxidative stress, senescence, AMPK, autophagy

## Abstract

Increased oxidative stress is a crucial factor for the progression of cellular senescence and aging. The present study aimed to investigate the effects of licochalcone D (Lico D) on oxidative stress-induced senescence, both in vitro and in vivo, and explore its potential mechanisms. Hydrogen peroxide (200 µM for double time) and D-galactose (D-Gal) (150 mg/kg) were used to induce oxidative stress in human bone marrow-mesenchymal stem cells (hBM-MSCs) and mice, respectively. We performed the SA-β-gal assay and evaluated the senescence markers, activation of AMPK, and autophagy. Lico D potentially reduced oxidative stress-induced senescence by upregulating AMPK-mediated activation of autophagy in hBM-MSCs. D-Gal treatment significantly increased the expression levels of senescence markers, such as p53 and p21, in the heart and hippocampal tissues, while this effect was reversed in the Lico D-treated animals. Furthermore, a significant increase in AMPK activation was observed in both tissues, while the activation of autophagy was only observed in the heart tissue. Interestingly, we found that Lico D significantly reduced the expression levels of the receptors for advanced glycation end products (RAGE) in the hippocampal tissue. Taken together, our findings highlight the antioxidant, anti-senescent, and cardioprotective effects of Lico D and suggest that the activation of AMPK and autophagy ameliorates the oxidative stress-induced senescence.

## 1. Introduction

Cellular senescence is a permanent condition of cell cycle arrest that promotes tissue remodeling during development, wound healing, host immunity, and tumor protection, but can also affect tissue homeostasis, tissue regenerative capacity, and inflammation, even leading to cancer [1,2]. Different types of cellular senescence have been identified based on its mechanism of activation and classified as either replicative senescence, physiological senescence, or stress-induced premature senescence [3]. Accumulation of senescent cells with age, one of the hallmarks of aging, contributes to the development of aging and aging-related diseases, including cancer, cardiovascular disease, and neurodegenerative diseases [4,5,6,7,8]. Conversely, clearance of these senescent cells can increase the health and lifespan of old mice [5]. Cellular senescence can occur naturally in vitro and in vivo and can be induced by cells or animals subjected to oxidative stress from substances such as hydrogen peroxide (H_2_O_2_) [9,10,11,12,13] and D-galactose (D-Gal) [14,15,16,17]. This type of senescence is called stress-induced premature senescence (SIPS). 

Hydrogen peroxide (H_2_O_2_) has been widely employed in vitro to generate oxidative stress and senescence as a result of apoptosis or the accumulation of apoptotic resistant senescence cells [13,18]. D-Gal treatment increases senescence in various organs, and heart and brain senescence have received the most attention [19,20]. Additionally, chronic administration of D-Galactose undergoes non-enzymatic glycation and react with amines to form Schiff’s base composite and Amadori products, which results in the increased level of AGE, RAGE and NADPH oxidase [20]. These factors cause oxidative stress and inflammatory pathway activation, which leads to cellular death and degenerative alterations, eventually leading to aging and age-related illnesses [21]. Therefore, it is necessary to develop a therapeutic strategy to either remove or inhibit the activation of these products under oxidative stress conditions.

Adenosine 5′ monophosphate-activated protein kinase (AMPK) is a cellular energy sensor that regulates a variety of functions including glucose, protein and lipid metabolism as well as autophagy and mitochondrial homeostasis [22]. It is widely known that increasing oxidative stress inhibits the expression of AMPK, whereas activation of AMPK has an anti-senescence impact and is implicated in autophagy induction [13,23,24,25]. Autophagy is a highly conserved catabolic process that maintains cellular homeostasis by degrading long-lived or dysfunctional proteins and organelle lysosomes of all living cells [26,27]. Previous studies have reported that AMPK activates autophagy directly (by phosphorylating ULK1 on at least four residues, Ser467, Ser555, Thr574, and Ser637) and indirectly (phosphorylates the mTOR upstream regulator TSC2 on Thr1227 and Ser1345 and the mTOR subunit RAPTOR on Ser722 and Ser792 by inhibiting mTORC1, which phosphorylates and inhibits ULK1) activating ULK1 [22,28,29]. Recent studies have demonstrated that autophagy is required to maintain stem cell properties, including proliferation, differentiation, and stem cell fate [28,29,30,31] and to prevent senescence both in vitro and in vivo under oxidative stress conditions [13,23,24,28,29,30,31]. Therefore, activation of AMPK could serve as a critical mediator of autophagy activation in response to protection from oxidative stress-induced senescence.

Licorice, also known as *Radix Glycyrrhizae*, is a perennial herb that has been used as an integral part of Chinese traditional medicine and Ayurveda for centuries. *Glycyrrhiza inflata* (*G. inflata*) is a major botanical source of licorice-containing species. *G. inflata* is chemically characterized and identified by the presence of retrochalcones, which are part of an important class of natural chalcones [32] and are precursors of flavonoids. The lack of oxygen functionality on C-2′ and C-6′structurally distinguishes these retrochalcones from other chalcones [33]. Based on their chemical structure, licochalcones are classified as licochalcones A, B, C, D, E, F, and G. Many biological activities have been reported for these chalcones, including anticancer, anti-inflammatory, antimicrobial, antiviral, antiallergic, antioxidant, osteogenic, and antidiabetic properties [34]. It has also been observed that licorice extract diminishes brain aging by inhibiting oxidative stress and neuronal apoptosis [35]. Although considerable evidence has shown that licorice has a remarkable anti-aging effect in senescence related models, little research has shown which components from the licorice provide these beneficial effects. 

In this study we chose Licochalcone D (Lico D) to explore its role in anti-aging properties. Lico D was first isolated as a prenylated retrochalcone from *Glycyrrhiza inflata (G. inflata)* in 1992 and its chemical structure was elucidated as 2-methoxy-3′prenyl-3,4,4′-trihydroxychalcone [33]. It possesses antioxidant, anti-inflammatory, anticancer, antiviral, and cardioprotective activities [21,22,23,24,25,26,27]. However, the effect of Lico D on senescence and aging remains unknown. Therefore, we investigated the effects of Lico D on senescence and aging by (1) testing the protective effects of Lico D against H_2_O_2_ induced oxidative stress in human bone marrow-mesenchymal stem cells (hBM-MSCs); (2) confirming the effect of Lico D on oxidative stress-induced senescence in vitro and in vivo; (3) monitoring the mechanism of action of Lico D in the senescence model. Our results indicate that Lico D protects cells and tissues from oxidative stress-induced senescence via the activation of AMPK.

## 2. Results

### 2.1. Lico D Protects the hBM-MSCs against Oxidative Stress

Initially, in order to determine the effect of Lico D on hBM-MSC viability, we treated hBM-MSCs with different concentrations of Lico D (0–8 µg/mL) for 12h and cell viability was measured. As shown in Figure 1A, there were no significant changes in cell viability upon Lico D treatment. However, previous reports highlighted that high concentration of Lico D possesses anti-cancer activity in different cancer cell lines [36,37,38]. Therefore, we chose the lowest concentration of Lico D (1 µg/mL) for our further in vitro experiments. The antioxidant properties of Lico D were evaluated against H_2_O_2_ in hBM-MSCs. Lico D-pretreated hBM-MSCs were exposed to different concentrations of H_2_O_2_ (0–1 mM) and cell viability was measured. The results indicated that cell viability was significantly increased in Lico D-pretreated hBM-MSCs (Figure 1B). Next, we examined the effect of Lico D on ROS generation and cell death. As expected, intracellular ROS levels were significantly increased upon H_2_O_2_ treatment, whereas they were significantly reduced in Lico D-pretreated hBM-MSCs (Figure 1C,D). Furthermore, these protective effects of Lico D were also confirmed by the expression levels of apoptotic proteins, such as p53 and cleaved caspase 3 (Figure 1E). Both markers were increased upon treatment with H_2_O_2_, whereas the expression of these markers was reduced in Lico D-pretreated hBM-MSCs. Collectively, these results suggest that Lico D protects hBM-MSCs against oxidative stress by inhibiting excess intracellular ROS production and cell death.

### 2.2. Lico D Reduces the Oxidative Stress Induced Senescence in hBM-MSCs

To investigate the protective effect of Lico D in hBM-MSC senescence, the cells were subjected to stress-induced premature senescence (SIPS), particularly the oxidative stress-induced senescence model, as described in the Methods section. As shown in Figure 2B, the oxidative stress-induced senescence cells became enlarged, flattened, and highly SA-β-gal-positive cells. The accumulation of SA-β-gal positive cells was significantly augmented in the double time H_2_O_2_ treated hBM-MSCs (40%) when compared with single time treated hBM-MSCs (12%) (Figure 2C). Therefore, we chose a double-time H_2_O_2_ treatment for further studies. As expected, the number of SA-β-gal-positive cells was significantly reduced by treatment with Lico D (13%) and ascorbic acid (17%, a positive control). Furthermore, the anti-senescence effects of Lico D were evaluated by the expression of well-known senescence markers, including p53, p16, and p21. As shown in Figure 3A–D, the expression of senescence markers was increased in double time H_2_O_2_ treated hBM-MSCs, while it was reduced in Lico D-treated hBM-MSCs. These results suggest that Lico D not only protects hBM-MCSs from oxidative stress, but also reduces oxidative stress-induced senescence in hBM-MSCs.

### 2.3. Lico D Reduces the Oxidative Stress Induced Senescence via Activation of AMPK and Autophagy

Next, we assessed the role of AMPK in our senescence model, as AMPK is a well-established mediator in the prevention of oxidative stress. The results showed that AMPK activation was significantly decreased in the oxidative stress-induced senescence model (Figure 3E,F), while AMPK activation was significantly restored in Lico D-treated senescent cells. Furthermore, this effect was confirmed using compound C (CC), a potential AMPK inhibitor. As expected, Lico D-mediated AMPK activation was abolished by treatment with CC (Figure 3E,F). These data suggest that AMPK activation by Lico D can reduce oxidative stress-induced senescence in hBM-MSCs, and this reduction could be prevented by CC. 

Further, we extended our study to explore the role of Lico D in autophagy activation. The results showed that Lico D treatment significantly increased the expression levels of LC3 and BECN1 and decreased the expression of SQSTM1 (Figure 4A–D). To verify the involvement of AMPK in autophagy, we used the CC. Interestingly, we found that AMPK-mediated autophagy was inhibited by treatment with CC. Taken together, our findings indicate that Lico D activates autophagy through AMPK and reduces oxidative stress-induced senescence. 

### 2.4. Effect of Lico D on Body Weight in D-Gal Induced Aging Mice

The animal’s body weight was measured every week until the experiment ended. No significant differences were observed in either control or D-Gal or Lico D-injected animals (Figure 5B).

### 2.5. Lico D Reduces RAGE Expression in D-Gal Induced Aging Mice

Long-term D-Gal treatment can cause cognitive dysfunction by causing oxidative stress and neuroinflammation, both of which are linked to signs of aging in the brain. To identify the role of Lico D in hippocampal inflammation, we screened the aging-related inflammation marker RAGE. Our data showed that D-Gal treatment increased the expression of RAGE at the mRNA level compared to that in the control group (Figure 5C). Treatment with Lico D significantly reduced RAGE expression compared to that in D-Gal-treated mice. Our results suggest that Lico D can reduce D-Gal-induced hippocampal inflammation by reducing RAGE expression. 

### 2.6. AMPK Activation by Lico D Ameliorates Heart and Hippocampus Senescence in D-Gal Induced Aging Mice

Oxidative stress can activate cellular senescence through various signaling cascades, but ultimately activate p53, p16, or both, and mediate cell cycle arrest [39,40]. Therefore, we first examined the expression of senescence markers p53 and p21 in the heart and hippocampus tissues. As expected, the expression levels of p53 and p21 were upregulated in D-Gal-injected mice compared to those in the vehicle (Figure 6A–C heart, and F-H hippocampus). More importantly, their expression was significantly reduced in the Lico D-treated group. These data suggest that Lico D ameliorates heart and hippocampal senescence in D-Gal-induced aging mice. Next, we analyzed the effect of Lico D on AMPK activation in the heart and hippocampus. There were no significant changes in the AMPK levels in the heart and hippocampus tissues of D-Gal-treated mice (Figure 6D,E hearts; and I and J hippocampus). However, when compared to the D-Gal treatment, AMPK activation was significantly increased in both tissues of Lico D. These results suggest that Lico D reduces senescence by activating AMPK in D-Gal-induced aging mice. 

### 2.7. Lico D Activates Autophagy in D-Gal Induced Aging Mice Heart Tissue

To confirm the effect of Lico D on autophagy activation, we analyzed the expression levels of autophagy markers in our mouse model. As shown in the Figure 7A–D autophagy activation was significantly decreased in D-Gal-treated mice heart tissue compared to that in the control group. Lico D administration markedly increased the expression of these autophagy markers in heart tissue. No differences were observed in hippocampal tissue (data not shown). Collectively, these results reinforce that Lico D reduces heart senescence by activating AMPK and autophagy in a D-Gal-induced aging mouse model. 

## 3. Discussion

In this study, we found that Lico D ameliorates senescence and aging, and obtained interesting findings: (i) Lico D reduced oxidative stress-induced premature senescence by AMPK activation both in vitro and in vivo; (ii) AMPK activation by Lico D can restore impaired autophagy under oxidative stress conditions; (iii) Lico D reduced RAGE expression in the hippocampal tissue. As AMPK is a master regulator of cellular metabolic homeostasis, our findings will be useful for further research on the role of AMPK activation in oxidative stress-induced senescence, which will hopefully contribute to reducing the burden of oxidative stress-induced cellular senescence during aging.

### 3.1. Lico D Ameliorated the Oxidative Stress Induced Premature Senescence via AMPK Activation

Owing to their multipotency, MSCs can be readily expanded and differentiated under appropriate condition in vitro and virtue of their trophic and growth factor-rich secretome, it can attract other cells necessary for tissue repair and help to create a regenerative environment in vivo [28,29]. Thus, MSCs are promising cell therapy candidates for regenerative medicine. In particular, the number of clinical trials has been growing in autologous stem cell therapy compared to allogenic applications [30]. However, the functionality and longevity of MSCs are reduced during ex vivo expansion, which is caused by increased levels of oxidative stress. Furthermore, these increased oxidative stresses may inhibit the proliferative and differentiation capacity of MSCs and direct them into senescence, resulting in reduced function and engraftment [31]. Therefore, finding a way to reduce the accumulation of oxidative stress may enhance the benefits of MSCs in regenerative medicine.

As reported earlier, Lico D possesses antioxidant effects against various oxidative stress conditions both in vitro and in vivo [41,42]. Therefore, we confirmed the antioxidant effects against H_2_O_2_ in hBM-MSCs. Interestingly, we found that the viability of Lico D-pretreated hBM-MSCs increased, ROS production and apoptotic cell death decreased under oxidative stress conditions (Figure 1B–E). Oxidative stress is a type of SIPS that can develop senescence faster than replicative senescence and has been extensively used to study the influence of extracellular and intracellular stress on the aging process. H_2_O_2_ and D-Gal are well-known inducers of oxidative stress in vitro and in vivo, respectively [13,14,15,16,17,43,44]. The present study was designed to assess the anti-senescence effect of Lico D in H_2_O_2_ induced senescence model in vitro and a D-Gal-induced senescence model in vivo. Due to the heterogeneous nature of senescent cells and the lack of specific markers, a combination of multiple techniques has been used to identify the senescent cells. In our study, SA-β-Gal staining and senescence markers, such as p21 and p53, were evaluated. Our findings confirmed that Lico D significantly reduced the accumulation of SA-β-Gal positive cells in vitro, as well as the reduction of the senescence markers p21 and p53 observed in both models (Figure 3 and Figure 6). 

AMPK activation plays an important role in cellular senescence and aging [45]. The AMPK pathway is downregulated in oxidative stress-induced senescence, whereas pharmacological activation of AMPK or swimming exercise training may increase AMPK and inhibit senescence-associated pathways [13,46]. Therefore, in our study, we attempted to determine the importance of AMPK under oxidative stress conditions. In our in vitro senescence model, AMPK activation was significantly reduced, whereas the reduction was greatly restored upon Lico D treatment (Figure 3E,F). To confirm the above finding, we used Compound C, a potent inhibitor of AMPK, and obtained consistent data with a previous report [13]. We used the D-Gal-induced aging model, which is a widely used artificial aging model, to obtain a better understanding of senescence and aging in vivo. Chronic administration of D-Gal induces degenerative changes in many tissues and organs, particularly in the heart and brain tissues were deeply focused [19,20,21]. Hence, we chose the heart and hippocampus tissues for our molecular experiments. Our results demonstrated that Lico D administration significantly upregulated AMPK in these tissues, whereas no changes were observed in D-Gal-treated mice (Figure 6D,I). Therefore, our findings strongly support the anti-senescence activity of Lico D against oxidative stress via AMPK activation. In general, AMPK activation is thought to be mediated by two distinct kinases, liver kinase B1 (LKB1) and calcium-sensitive kinase CAMKK2, as well as other factors such as allosteric activators (e.g., A-769662, salicylate, etc.), energy status modifiers (starvation, exercise, mitochondrial poisons like metformin), and AMP mimetics (AICAR) [22]. However, it remains unknown whether Lico D activates AMPK by one of the mechanisms mentioned above or by directly enhancing ROS scavenging activity and thus preventing AMPK inactivation. Considering this, activation of AMPK by Lico D requires further investigation.

### 3.2. Lico D Restored the Impaired Autophagy via AMPK Activation under Oxidative Stress

The anti-senescence effect of AMPK was closely associated with the induction of autophagy, which was determined by autophagic flux. Impairment of autophagic flux with lysosomal dysfunction was also observed in oxidative stress-induced senescence or H_2_O_2_ induced senescence [13,25]. In our results, we showed that Lico D treatment improved autophagy activation in our senescence model, which was detected by elevated levels of LC3 and BECN1 and reduction of SQSTM1. Furthermore, the importance of AMPK in autophagy was also evaluated using Compound C, which supported the hypothesis that Lico D activates autophagy by activating AMPK in our senescence model. Most importantly, we observed a similar effect in D-Gal-induced aging mouse heart tissue (Figure 7), but not in the hippocampus. One of the possible mechanisms behind this effect is that AMPK-mediated RAGE inhibition may trigger the anti-senescence effect of Lico D in hippocampal tissue. Collectively, AMPK facilitated autophagy may be one of the mechanisms by Lico D, which can reduce oxidative stress-induced premature senescence in vitro and in vivo. 

### 3.3. Lico D Reduced the RAGE Expression in Hippocampus

RAGE, the receptor for advanced glycation end product, was first characterized as a transmembrane receptor of the immunoglobulin superfamily in 1992 [47,48]. AGE activation is considered a major mediator of AGE pathogenicity, and AGE-RAGE association is the most studied phenomenon for the induction of oxidation and inflammation [49]. Therefore, targeted pharmacological interventions that can either inhibit or modulate the expression levels of AGE and RAGE, and their signaling could be a promising therapeutic strategy to slow down the development of aging and age-related diseases, including Alzheimer’s disease [50,51]. Given the importance of RAGE in this process, we evaluated RAGE expression in hippocampal tissues. Surprisingly, the expression level of RAGE was reduced in the Lico D-treated mice (Figure 5C). Consistent with previous reports, activation of AMPK could inhibit the AGE-induced inflammatory response and suppress RAGE/NFkB signaling [52,53,54,55,56]. Therefore, our findings suggest that Lico D-mediated AMPK activation is sufficient to reduce D-Gal-induced hippocampal RAGE expression. Overall, these results indicate that Lico D diminishes RAGE-mediated inflammation in D-Gal-induced aging mouse hippocampal tissue.

In summary, using an oxidative stress-induced senescence model, we were able to provide evidence that the effect of Lico D on senescence and aging relies on both the activation of AMPK and autophagy and the inhibition of RAGE mediated inflammation. Therefore, AMPK activation may involve multiple mechanisms in the cells to prevent oxidative stress-induced senescence and aging. To our knowledge, this is the first report that Lico D activates AMPK and autophagy to reduce oxidative stress-induced senescence (Figure 8). These findings suggest that Lico D may be a possible therapeutic candidate for the prevention or treatment of age-related diseases.

## 4. Materials and Methods

### 4.1. Chemicals and Reagents

Methylthiazolyldiphenyl-tetrazolium bromide (#M2128 MTT) assay; 2′,7′-dichlorofluorescin diacetate (#D6883), ascorbic acid (#A8960), H_2_O_2_ (#H1009), and D-Gal (#G0750) were purchased from Sigma-Aldrich (USA). S-β-gal staining assay kit (#9860) was obtained from Cell Signaling Technology (Danvers, MA, USA). Lico D (#CFN99805; purity ≥ 98%) was purchased from ChemFaces (430056; Wuhan, Hubei) and Compound C was purchased from Calbiochem (Darmstadt, Germany). Radioimmunoprecipitation (RIPA) lysis buffer (Santa Cruz Biotechnology, Dallas, TX, USA) and Pierce BCA protein assay kit and Hoechst 33342 Trihydrochloride Trihydrate (#H3570) were purchased from Thermo Fisher Scientific, USA, respectively. RNAiso Plus (#9109; Total RNA extraction reagent) and Primescript^TM^ II 1st strand cDNA synthesis kits (#6210A) were purchased from Takara Bio Inc. (Japan). Primary antibodies including p16 (#sc-1661), p21 (#sc-397), p53 (#sc-6243), Caspase-3 (#sc-7148); MAP LC3α/β (#sc-398822); BECN1 (#sc-48341); SQSTM1 (#sc-48402) and GAPDH (#sc-365062) were acquired from Santa Cruz Biotechnology, INC. (TX, USA). The anti-caspase 3 active form (#AB3623; 1:500) was purchased from Merck Millipore, Germany and AMPKα (#5832) and p-AMPK (#2535) were purchased from Cell Signaling Technology (USA). The appropriate HRP-conjugated secondary antibodies, mouse anti-rabbit (#sc-2357), and mouse anti-goat (#sc-2354) antibodies were purchased from Santa Cruz Biotechnology, Inc. (Dallas, TX, USA) and the horse anti-mouse (#7076) antibody was from Cell Signaling Technology (USA). ECL Western blotting detection reagents (RPN2209) were purchased from GE Healthcare (Buckinghamshire, UK).

### 4.2. hBM-MSCs Cell Culture

hBM-MSCs were acquired from Cell Engineering for Origin (CEFO), Seoul, Korea. The cells were free from bacterial, viral, and mycoplasmal contamination. The cells were characterized by flow cytometry analysis, which revealed the CD73+, CD105+, and CD31- phenotype (data not shown). The cells were grown in Dulbecco’s modified Eagle medium (DMEM) (Gibco, Life Technologies, Grand Island, NY, USA) supplemented with 10% FBS (Gibco, Life Technologies, USA), L-glutamine, and 1% penicillin/streptomycin solution (Lonza, Walkersville, MD, USA). The cells were manipulated under sterile conditions in a humidified incubator at 37 °C with 5% CO_2_ and% air. The cells were subcultured as soon as they reached confluence. The cells were serially monitored under bright field microscopy (Nikon Eclipse TS100; Tokyo, Japan), and the media was changed every 3 d. 

### 4.3. Cell Viability Assay

The cytotoxic and protective effects of Lico D in hBM-MSCs were determined using the methylthiazolyldiphenyl-tetrazolium bromide (MTT) test, according to the manufacturer’s instructions. To check the cytotoxic effect, hBM-MSCs were grown in a 96-well plate at a density of 8 × 10^3^ cells per well. The next day, the cells were treated with different concentrations of Lico D (1–8 µg/mL) for 12 h, followed by incubation with MTT solution for 2 h, and the formazan crystals were dissolved using dimethyl sulfoxide (DMSO). Cell viability was assessed using a spectrophotometer (Multiskan FC, Thermo Fisher Scientific). For the protective effect of Lico D, cells were pre-treated with 1 µg/mL of Lico D for 12 h followed by incubation with different concentrations of H_2_O_2_ (0.5–1 mM) for 1 h. Following H_2_O_2_ treatment, cells were treated with MTT solution for 2h, and cell viability was evaluated. 

### 4.4. Detection of Intracellular ROS

To quantify the production of intracellular ROS, we used cell permeable substrate 2′,7′-dichlorofluorescin diacetate (DCFH-DA), which can be converted to highly detectable fluorescent 2′,7′-dichlorofluorescein upon oxidation. The cells were cultured in 12-well plates and 96-well plates for 24 h. After 24 h of culture, the cells were treated with 1 µg/mL of Lico D for 12 h and incubated with 20 μM DCFH-DA for 30 min. The cells were then washed and incubated with H_2_O_2_ (0.7 mM) for 1 h at 37 °C. The cells were then fixed with 4% paraformaldehyde at room temperature for 15 min and the nuclei were stained with Hoechst. Then, the cells were observed under a fluorescent microscope (Nikon Eclipse Ti2; Japan), and the images were captured (Nikon DS-Ri2; Japan) and analyzed using the Image J software. To analyze the apoptotic markers, the cells were pretreated with 1 µg/mL of Lico D followed by incubation with H_2_O_2_ (0.7 mM) for 1 h at 37 °C, and then the proteins were isolated as described below.

### 4.5. Oxidative Stress-Induced Senescence

H_2_O_2_ was used to induce oxidative stress-induced cell cycle arrest and senescence as described previously [9,44] with some modifications (Figure 2A). Briefly, the cells were exposed to 200 µM H_2_O_2_ for 2 h and cultured for 2 d without H_2_O_2_. In the second treatment with H_2_O_2_, the cells were split 1:3, exposed to 200 µM H_2_O_2_ for 2 h, and cultured with either normal media (control), Lico D (1 µg/mL), ascorbic acid (500 µM AA; a positive control), or the compound C (0.5 µM; AMPK inhibitor) for 72 h. Cellular senescence was confirmed by SA-β-gal assay and Western blotting as described below. 

### 4.6. Senescence-Associated β-Galactosidase (SA-β-gal) Staining

Senescence-associated β-galactosidase positive cells were identified using an SA-β-gal assay kit (Cell Signaling Technology, Danvers, MA, USA) according to the manufacturer’s instructions. Briefly, after 72 h of treatment, cells were washed with phosphate-buffered saline (PBS) and fixed with 1X fixative for 10–15 min at room temperature. The β-galactosidase staining solution (pH 6.0) was added to the plates and incubated overnight in a dry incubator at 37 °C without CO_2_. The next day, the β-Gal positive cells were observed under a light microscope (Nikon Eclipse TS100; Tokyo, Japan) and captured using a Canon i-Solution IMTcam3 digital camera (Tokyo, Japan). 

### 4.7. Immunoblotting Analysis

Total proteins were extracted with RIPA lysis buffer system containing phenylmethylsulfonyl fluoride (PMSF), sodium orthovanadate (Na3VO4), and protease inhibitor cocktail (Santa Cruz Biotechnology) at 4 °C for 30 min, and the samples were centrifuged at 16,000× *g* for 20 min. The total protein concentration was quantified using the Pierce BCA Protein Assay Kit (Thermo Fisher Scientific). Proteins (30–50 µg) were loaded and separated via sodium dodecyl sulphate–polyacrylamide gel electrophoresis (SDS-PAGE) gel electrophoresis, followed by blotting on a PVDF membrane (GE Healthcare, Germany). The membranes were blocked with 1% blocking solution containing non-fat dry milk in TBST for 1 h 30 min at RT to prevent non-specific binding of primary antibodies. Subsequently, the membranes were incubated overnight at 4 °C with the corresponding primary antibodies. Secondary antibodies were conjugated with horseradish peroxidase and visualized using an enhanced chemiluminescence detection kit (GE Healthcare). Densitometry analysis was performed using the ImageJ software.

### 4.8. Animals and Administration of Drugs

Six-week-old male C57BL/6 mice (weighing 22 ± 2 g) were purchased from Samtako Bio Korea Co., Ltd. (Osan, Gyeonggi, Korea) and maintained at 23–25 °C under a 12-h light and 12-h dark cycle with free access to food and water in a pathogen-free facility. All animal experiments were approved by the Institutional Animal Care and Use Committee of Chosun University (CIACUC2020-A0009). After a week of acclimation, the animals were randomly divided into three groups (four mice per group): normal control group (PBS alone), D-Gal model group (150 mg/kg/day), and D-Gal + Lico D (0.5 mg/kg/day). The mice were intraperitoneally injected with either PBS or D-Gal for 10 weeks and from the third week onwards, the mice were intraperitoneally injected with Lico D for 8 weeks (Figure 5A). Body weight was measured every week until the end of the experiment.

### 4.9. RNA Extraction and Quantitative Reverse Transcription-Polymerase Chain Reaction (qRT-PCR)

RNAisoPlus (Takara) was used to isolate total RNA from the hippocampus and heart tissues. Total RNA (2.5 μg) was then reverse-transcribed using Primescript^TM^ II 1st strand cDNA synthesis kit (Takara) and quantified using the Power SYBR Green PCR Master mix (Applied BioSystems). The following mouse primers were used to amplify the β-actin forward primer 5′-CCACCATGTACCCAGGCATT-3′ and reverse primer 5′-CGGACTCATCGTACTCCTGC-3′ and RAGE—forward primer 5′-AGGTGGGGACATGTGTGTC-3′ and reverse primer 5′-TCTCAGGGTGTCTCCTGGTC-3′. Real-time PCR reactions were performed using the StepOne^TM^ Real-Time PCR system (Applied Bio Systems) and the primer pairs were synthesized by GenoTech (Daejeon, Korea) or IDT (Integrated DNA Technologies, Coralville, IA, USA).

### 4.10. Statistical Analyses

All data are presented as the mean ± standard deviation from at least three or more biological replicates. The differences between data sets were assessed by Student’s t-test and analysis of variance (ANOVA) with Holm-Sidak’s multiple comparison test using GraphPad Prism (GraphPad Software). Statistical significance levels are indicated in the figures using asterisks as follows: * *p* < 0.05, ** *p* < 0.01, *** *p* < 0.001, and **** *p* < 0.0001.

## Figures and Tables

**Figure 1 ijms-22-07324-f001:**
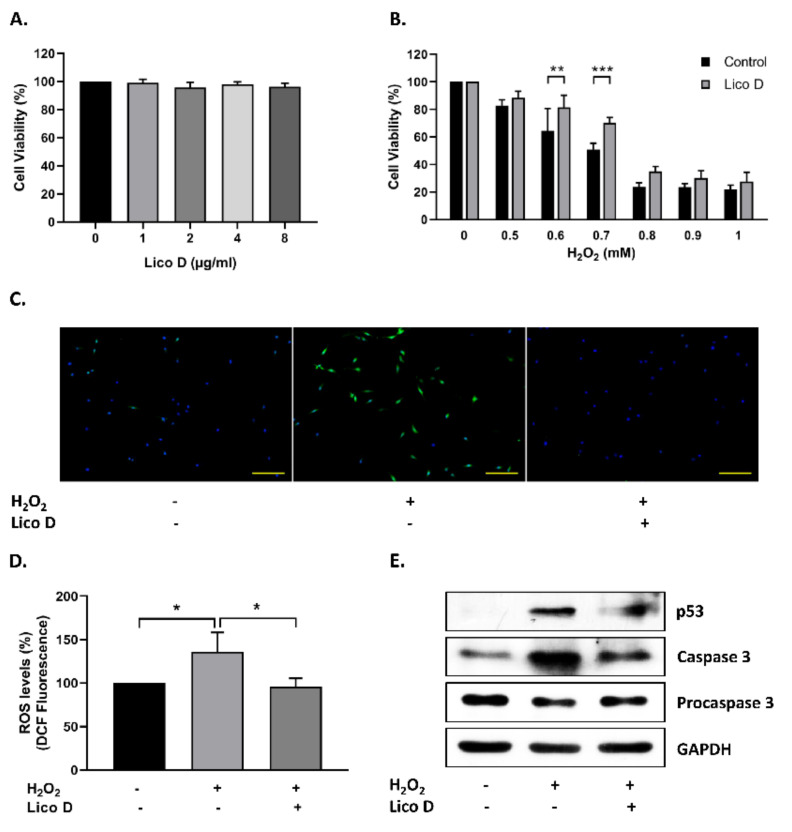
Protective effects of Licochalcone D (Lico D) against hydrogen peroxide (H_2_O_2_). (**A**) Effect of Lico D on the cell viability was determined by MTT assay. (**B**) Protective effect of Lico D against H_2_O_2_ was confirmed by MTT assay. (**C**) ROS scavenging activity of Lico D was determined by DCFH-DA fluorescence assay and (**D**) fluorescence intensity was quantified using Image J software. (**E**) Representative images from immunoblot assay against apoptotic markers such as procaspase 3, Caspase 3 (cleaved form) and p53. GAPDH was used as an internal control. The scale bar represents 100 µm. All data are represented as the mean ± standard deviation (SD) (*n* = 3) * *p* < 0.05, ** *p* < 0.01, and *** *p* < 0.001.

**Figure 2 ijms-22-07324-f002:**
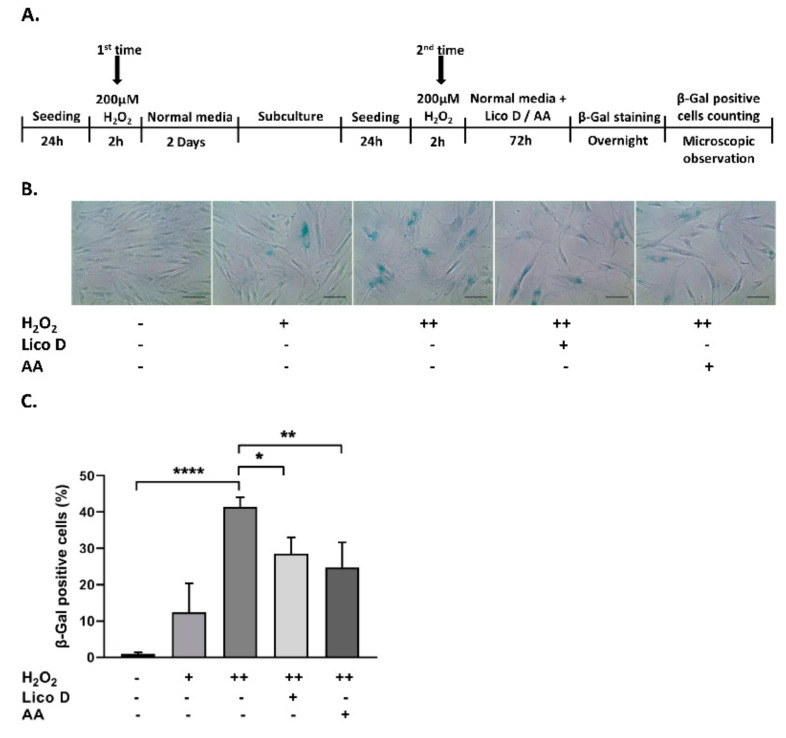
Effect of Lico D in oxidative stress induced senescent hBM-MSCs. (**A**) Graphical representation of the oxidative stress induced senescence in hBM-MSCs. (**B**) Representative image of SA-β-gal positive cells and (**C**) percentage of SA-β-gal positive cells in oxidative stress induced senescence model. H_2_O_2_ (+) represents one-time treatment and H_2_O_2_ (++) represents double time treatment. The scale bar represents 100 µm. All data are represented as mean ± SD (*n* = 3) * *p* < 0.05, ** *p* < 0.01, and **** *p* < 0.0001.

**Figure 3 ijms-22-07324-f003:**
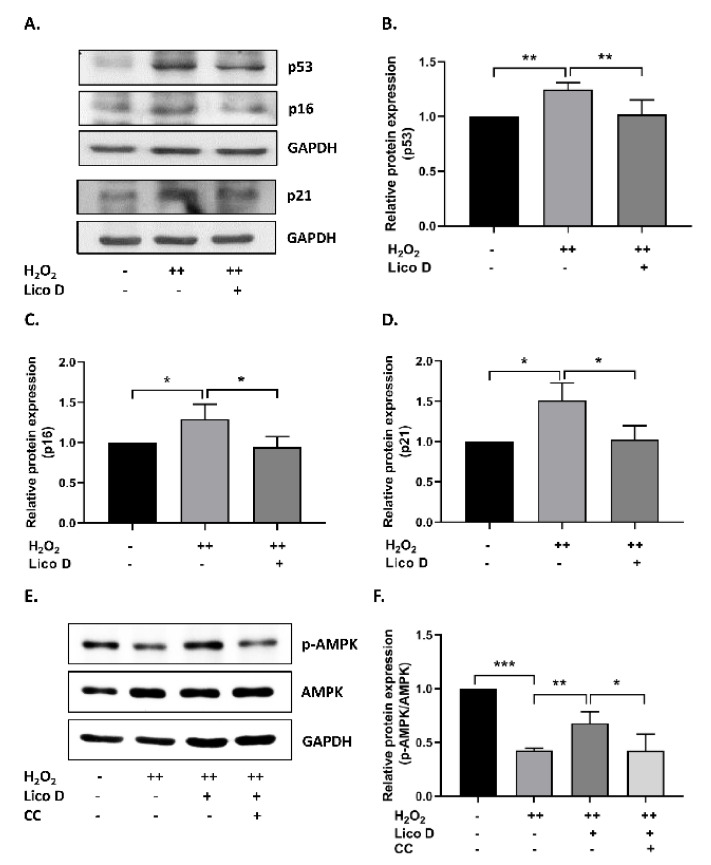
Lico D reduce senescence via AMPK activation in hBM-MSCs. (**A**) Representative images from immunoblot against senescence markers p53, p16, and p21 and (**B**–**D**) the expression level of target proteins was quantified using Image J software. (**E**) Representative images from immunoblot assay against AMPK, and phosphorylated AMPK and (**F**) the expression level of target proteins was quantified using Image J software. GAPDH was used as an internal control. Compound C (CC), an AMPK inhibitor was used. H_2_O_2_ (+) represents one-time treatment and H_2_O_2_ (++) represents double time treatment. The scale bar represents 100 µm. All data are represented as mean ± SD (*n* = 3) * *p* < 0.05, ** *p* < 0.01, and *** *p* < 0.001.

**Figure 4 ijms-22-07324-f004:**
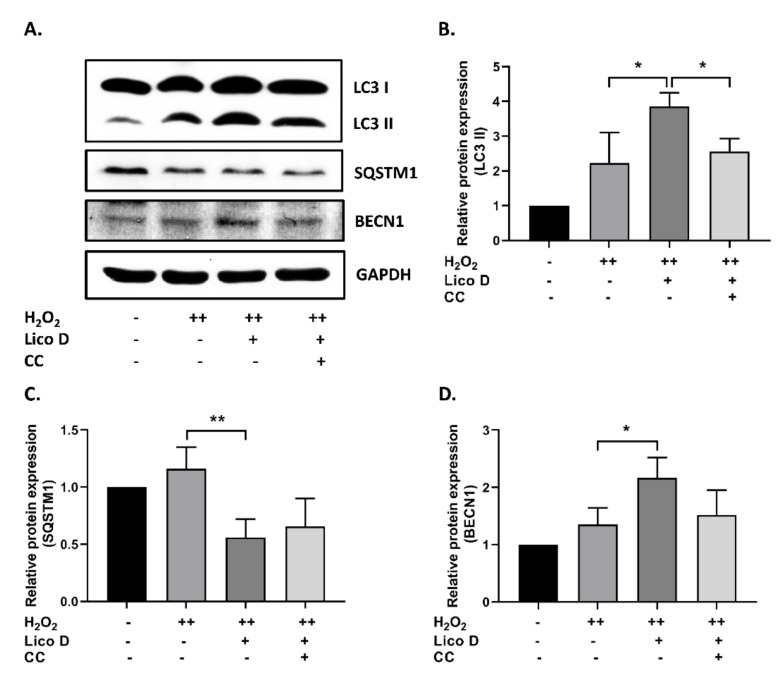
AMPK activation increases the autophagy in oxidative stress induced senescent hBM-MSCs. (**A**) Representative images from immunoblot against autophagy markers including LC3 II, BECN1, and SQSTM1. (**B**–**D**) The expression level of target proteins was quantified using Image J software. GAPDH was used as an internal control. Compound C (CC), an AMPK inhibitor was used. The results are represented as mean ± SD (*n* = 3). * *p* < 0.05 and ** *p* < 0.01.

**Figure 5 ijms-22-07324-f005:**
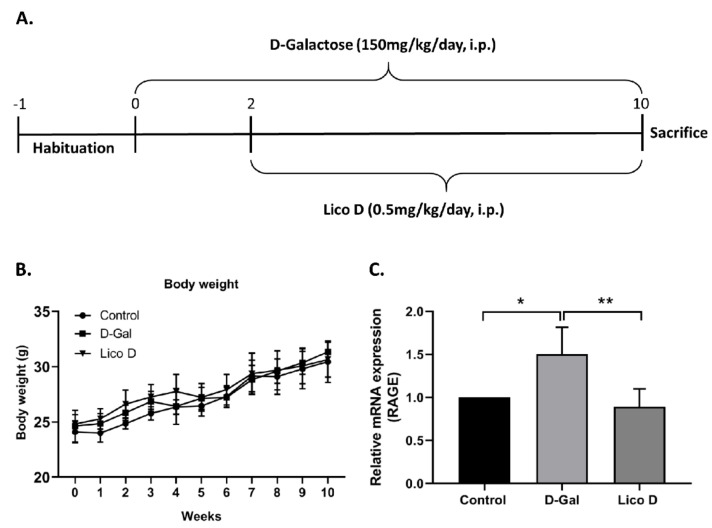
Lico D reduces receptors for advanced glycation end products (RAGE) expression in D-Galactose treated mice. (**A**) Schematic representation of the experimental plan and (**B**) body weights of the mice. (**C**) Relative fold changes in mRNA of RAGE was quantified by quantitative reverse transcription-polymerase chain reaction (qRT-PCR). All data are represented as mean ± SD (*n* = 4). * *p* < 0.05 and ** *p* < 0.01.

**Figure 6 ijms-22-07324-f006:**
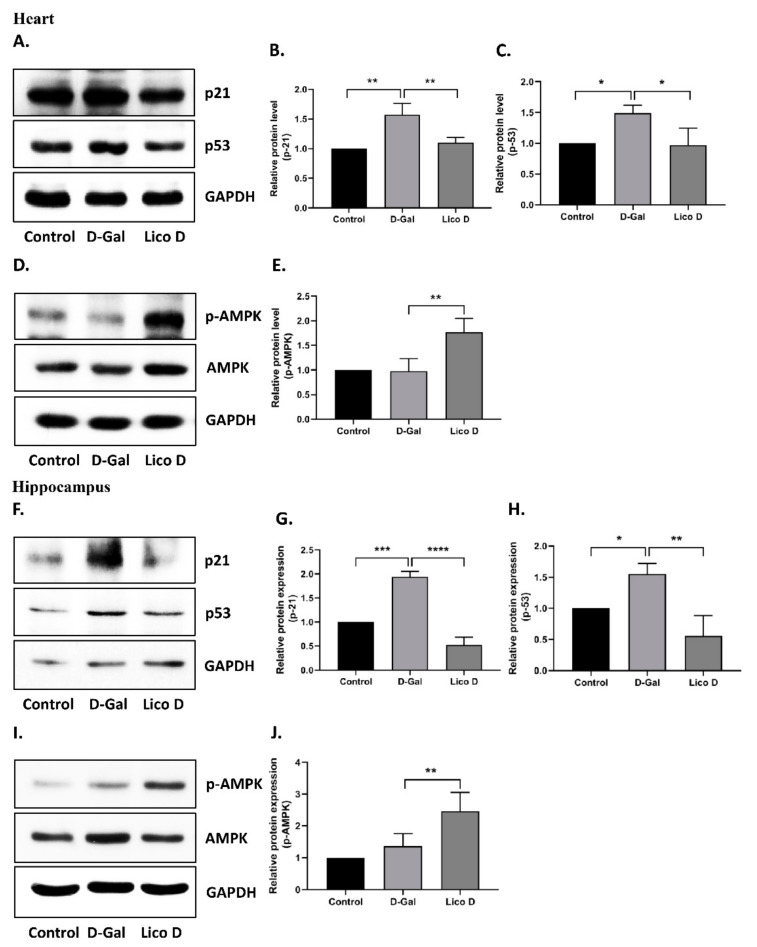
AMPK activation ameliorates the senescence in heart and hippocampus. Representative images from immunoblot against p53, p21 and GAPDH from heart (**A**–**C**) and hippocampus (**F**–**H**) tissues. The representative images from immunoblot against AMPK, p-AMPK and GAPDH from heart (**D**,**E**) and hippocampus (**I**,**J**) tissues. The expression levels of proteins were quantified using Images J software. All data are represented as mean ± SD (*n* = 3–4) * *p* < 0.05; ** *p* < 0.01; *** *p* < 0.001 and **** *p* < 0.0001.

**Figure 7 ijms-22-07324-f007:**
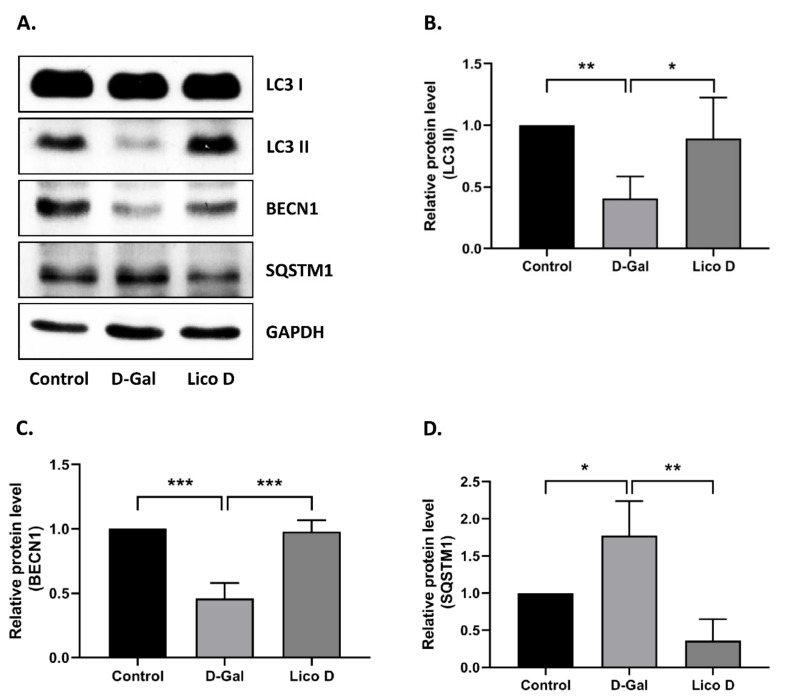
AMPK activation improved autophagy in D-Gal induced aging mice heart tissue. Representative images from immunoblot against LC3 II, BECN1, SQSTM1 and GAPDH from heart (**A**–**D**) tissue. The expression levels of proteins were quantified using Images J software. All data are represented as mean ± SD (*n* = 3). * *p* < 0.05, ** *p* < 0.01, and *** *p* < 0.001.

**Figure 8 ijms-22-07324-f008:**
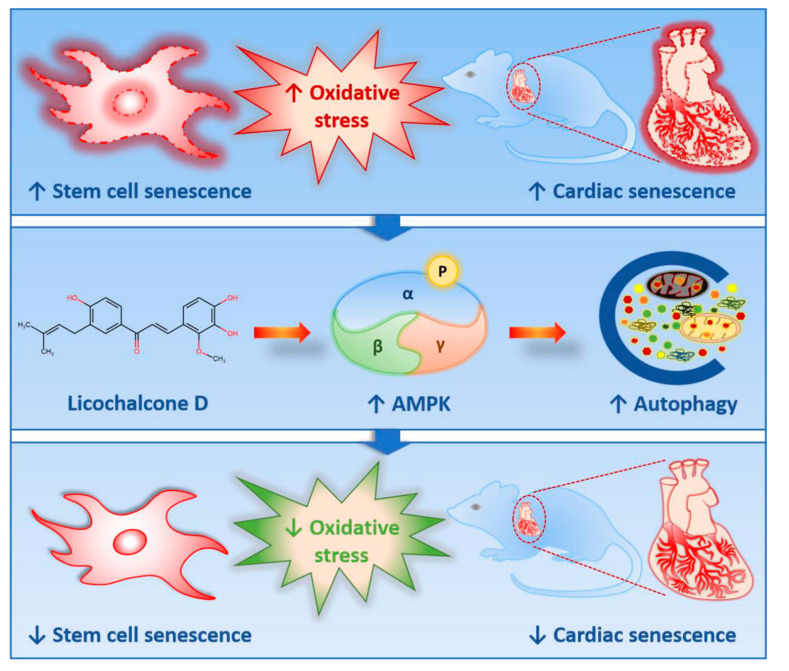
Lico D ameliorates oxidative stress induced senescence in stem cells and aging mice via AMPK/autophagy.

## Data Availability

The data presented in this study are available on request from the corresponding author.

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
