# Peer review of "Licochalcone D Ameliorates Oxidative Stress-Induced Senescence via AMPK Activation"

_ijms, 2021, doi:10.3390/ijms22147324_

Round 1

Reviewer 1 Report

The conducted research is very interesting, although the presentation of the obtained results is completely unreadable. The authors have at their disposal the so-called a discussion where they can relate to the results of other research groups. Placing descriptions of other authors' results in the results section makes their own results completely illegible. The introduction is not introductory to the research, the authors describe AMPK, Licochalcone and then refer to RAGE in their conclusions. The introduction needs to be changed. As for the discussion, I will speak out after correcting the introduction and the results section

Reviewer 2 Report

In this work, Nagarajan Maharajan et al. studied the effects of a known antioxidant, Lico D from a Chinese herb, on oxidative stress in a cell model and an animal model. The results convincingly indicated that AMPK activity, which is an established mediator in preventing oxidative stress induced cell death, is restored by treatment of Lico D, providing a molecular mechanism for its antioxidant effects. Overall, this is a piece of interesting and useful work. One issue which is still unclear is how Lico D restores AMPK activity. It is conceivable that this compound may directly scavenge ROS to exert the antioxidation function and thus prevent AMPK inactivation. Alternatively, this compound may directly take actions on the kinase to increase activity. A discussion of potential working mechanism(s) should be included. Also, there are many gramma mistakes throughout the manuscript, which need to carefully corrected.

Round 2

Reviewer 1 Report

The article may already be published in this form. The authors responded to my comments.